# An Analysis of Primary Hyperparathyroidism in Individuals Diagnosed with Multiple Endocrine Neoplasia Type 2

**DOI:** 10.3390/diseases13040098

**Published:** 2025-03-27

**Authors:** Ana-Maria Gheorghe, Claudiu Nistor, Alexandru-Florin Florescu, Mara Carsote

**Affiliations:** 1PhD Doctoral School of “Carol Davila” University of Medicine and Pharmacy, 020021 Bucharest, Romania; ana-maria.gheorghe@drd.umfcd.ro; 2Department 4-Cardio-Thoracic Pathology, Thoracic Surgery II Discipline, “Carol Davila” University of Medicine and Pharmacy, 050474 Bucharest, Romania; 3Thoracic Surgery Department, “Dr. Carol Davila” Central Military University Emergency Hospital, 010242 Bucharest, Romania; 4Endocrinology Department, “Grigore T. Popa” University of Medicine and Pharmacy, 700111 Iasi, Romania; 5Endocrinology Department, “Sf. Spiridon” Emergency County Clinical Hospital, 700111 Iasi, Romania; 6Department of Endocrinology, “Carol Davila” University of Medicine and Pharmacy, 020021 Bucharest, Romania; carsote_m@hotmail.com; 7Department of Clinical Endocrinology V, “C.I. Parhon” National Institute of Endocrinology, 011863 Bucharest, Romania

**Keywords:** parathyroid, parathormone, primary hyperparathyroidism, calcium, surgery, gene, multiple endocrine neoplasia, MEN2, MEN2A, RET

## Abstract

**Background:** Primary hyperparathyroidism (PHPT) represents a multi-faced disease with a wide spectrum of manifestations. Familial forms of PHPT (affecting up to 10% of the cases) involve a particular category that encompasses a large range of hereditary syndromes, including parathyroid hyper-function, frequently in the setting of a multi-glandular disease. **Objective**: The aim was to analyze the most recent findings regarding PHPT in multiple endocrine neoplasia type 2 (MEN2) to a better understanding of the timing with respect to the associated ailments, MEN2-related PHPT (MEN2-PHPT) clinical and genetic particularities, optimum diagnostic, and overall management, particularly, surgical outcomes. **Methods:** This was a PubMed-based compressive review with regard to the latest data published in English from January 2020 until January 2025, using the following keywords: “primary hyperparathyroidism” and “multiple endocrine neoplasia”, “multiple endocrine neoplasia type 2”, “MEN2”, or “MEN2A”. We included original full-length studies of any study design that provided clinically relevant data in MEN2-PHPT and excluded reviews, meta-analysis, and case reports/series. **Results:** A total of 3783 individuals confirmed with MEN2 or *RET* pathogenic variants carriers were analyzed across 14 studies that provided data on PHPT. The prevalence of MEN2-PHPT subjects varied between 7.84% and 31.3%, with particularly low rates in non-index patients (3.8%). PHPT was the first syndrome manifestation in 0.9% of MEN2 patients. In terms of gender distribution, females represented 42.85% or 54.9% (similar rates between women and men, and only a single cohort showed a female rate up to 80%). Most subjects were diagnosed with PHPT and underwent surgery in the third or fourth decade of life. The highest median age at MEN2 diagnosis was 42 years. The youngest patients were *RET* pathogenic variants carriers who underwent (genetic) screening with median ages of 12 or 14 years. *RET* pathogenic variants analysis (n = 10/14 studies) showed that 16.67% of patients with p.Cys634Arg and 37.5% of those with p.Cys611Tyr had symptomatic PHPT, while those with p.Cys618Phe and p.Leu790Phe were asymptomatic. Timing analysis with respect to the medullary thyroid carcinoma diagnosis showed synchronous PHPT diagnosis in 80% and metachronous in 10% of MEN2 patients; with respect to MEN2-pheochromocytoma, synchronous diagnosis of PHPT was found in 56%, while pheochromocytoma was identified before PHPT in 22% of the cases and after PHPT in 22%. Studies (n = 10/14, N = 156 subjects with MEN2-PHPT) on parathyroidectomy identified that 72.7% to 100% of the individuals underwent surgery, typically performed in adulthood, at ages spanning from a mean of 34.7 to 48.5 years. The post-surgery outcomes varied (e.g., the rate for persistent PHPT was of 0%, 8% to 16.7%; recurrent PHPT of 12.5% to 23%; permanent hypoparathyroidism of 33% to 46%; permanent unilateral vocal cord palsy of 0% up to16.7%). Data regarding the number of involved glands (n = 7, N = 77): the prevalence of multi-glandular disease was pinpointed between 12.5% and 50%. **Conclusions**: MEN2-PHPT involved unexpected high rates of single-gland involvement (from 33.3% to 87.5%), probably due to an early detection across genetic screening. Traditional female higher prevalence in PHPT was not confirmed in most MEN2 cohorts. As expected, a younger age at PHPT diagnosis and surgery than seen in non-MEN2 patients was identified, being tidily connected with the syndromic constellation of tumors/malignancies. Overall, approximately, one out of ten patients were further confirmed with MEN2 starting with PHPT as the first clinically manifested element.

## 1. Introduction

Primary hyperparathyroidism (PHPT) represents a multi-faced disease with a wide spectrum of causes, manifestations, and recently recognized forms such as asymptomatic and normocalcemic PHPT [1,2,3]. Familial forms of PHPT involve a particular category of this disease and encompass a large range of hereditary syndromes that predisposes individuals to parathyroid hyper-function, most frequently in the setting of a multi-glandular disease in addition to other endocrine and non-endocrine ailments [4,5,6,7]. Based on the underlying genetic defect, there are various syndromes, such as multiple endocrine neoplasia type 1 (MEN1) caused by pathogenic variants of *MEN1* gene, multiple endocrine neoplasia type 2 (MEN2) due to pathogenic variants of *RET* oncogene, multiple endocrine neoplasia type 4 (MEN4) harboring *CDKN1B*/p27 defects, hyperparathyroidism-jaw tumor syndrome due to pathogenic variants of *CDC73* gene, as well as familial isolated primary hyperparathyroidism [8,9,10,11,12].

MEN2, an autosomal dominant syndrome [13], used to be further classified as MEN2A and MEN2B. However, more recently, the “MEN2” term was adopted for MEN2A, while MEN2B became known as “MEN3” [14]. *RET*, a gene located on chromosome 10q11.2 which encodes a tyrosine kinase receptor, is related to cellular proliferation, differentiation, and survival [15,16]. Tumorigenesis in MEN2 typically involves medullary thyroid carcinoma (MTC), as the key finding, associated with pheochromocytoma (PC) and/or PHPT, the endocrine manifestation which presents the lowest prevalence [17,18,19,20].

PHPT may be either by a single-glandular or multi-glandular disease, previously classified as adenoma and hyperplasia [21]. It affects up to one third of individuals confirmed with MEN2, and it may be symptomatic (causing nephrolithiasis/renal failure, osteoporosis and low trauma fractures, gastrointestinal disturbances, cardio-metabolic issues, as well as neuropsychiatric symptoms that overall requires a multidisciplinary team), or it may be identified in asymptomatic patients, through calcium testing or genetic screening protocols [22,23,24,25,26,27]. The single definitive management of PHPT is parathyroidectomy (PTx). The surgical strategy is complex and is based on the number of hyper-functioning glands, therefore imperiously needing preoperative localization via imaging studies such as neck ultrasound, Tc-99m-sesta-Methoxy isobutyl isonitrile scintigraphy, computed tomography, and single-photon emission computed tomography [28,29,30]. PTx may be focused (e.g., for a single gland), subtotal, and minimally invasive or even total with auto-transplantation to a heterotopic site such as the forearm [31]. Another important aspect is the concomitant presence of other endocrine neoplasms/malignancies, and the high probability that PTx is practiced on a previously explored neck for MTC [32]. Due to younger age at presentation, multi-glandular involvement and inheritance risk, a tailored management approach is needed, including (family) genetic counselling, refined screening methods, and a personalized surgical approach provided by a skillful team [33,34].

The aim was to analyze the most recent findings regarding PHPT in MEN2 patients for a better understanding of the timing with respect to the associated ailments amid the syndrome, MEN2-related PHPT clinical and genetic particularities, optimum diagnostic, and overall management, particularly, surgical outcome.

## 2. Methods

This was a PubMed-based compressive review (of narrative design) with regard to the latest data published in English from January 2020 until January 2025, using the following keywords in different combinations: “primary hyperparathyroidism” and “multiple endocrine neoplasia”, “multiple endocrine neoplasia type 2”, “MEN2”, “MEN2A”. We only included original, full-length studies of any study design that provided clinically relevant data in MEN2-related PHPT and excluded reviews, meta-analysis and case reports/series. A total of 14 papers were finally analyzed [35,36,37,38,39,40,41,42,43,44,45,46,47,48] (Figure 1).

## 3. Results

The 14 studies [35,36,37,38,39,40,41,42,43,44,45,46,47,48] reporting data about PHPT in patients with MEN2 included 218 subjects with PHPT and MEN2, out of a total 3783 individuals confirmed with MEN2 or *RET* pathogenic variants carriers (N = 7963 subjects across the entire studied population) [35,36,37,38,39,40,41,42,43,44,45,46,47,48] (Table 1).

The prevalence of PHPT in MEN2 subjects varied between 7.84% [38] and 31.3% [45], with particularly low prevalence in non-index-patients (3.8%) [39]. Apart from the differences in terms of prevalence observed between index and non-index cases (9% versus 3.8%, *p* = 0.019) [39], subjects with high-risk pathogenic variants also had a higher prevalence of PHPT compared with moderate-high and low-risk pathogenic variants (11.4% versus 2.4% versus 0.5%, *p* < 0.001) [40].

PHPT was the first manifestation in 0.9% of patients with MEN2 [47]. MEN2 in subjects confirmed with PHPT was found in 0.12% of them [37], 4.8% [42], and 22.9% [36]. In terms of female-to-male ratio, one cohort reported that 54.9% of the patients with MEN2-related PHPT were females [40], while another found a rate of 42.85%, with no statistically significant difference between males and females (*p* = 0.090) [43]; another study reported a female prevalence of 80% in MEN2-associated PHPT [37]. Moreover, 80% of PHPT subjects with the parathyroid condition as the first MEN2 manifestation were females [47].

Most patients were diagnosed with PHPT and underwent PTx in the third [35,37,40,47,48] or fourth [36,38,39,40] decade of life. The highest median age at MEN2 diagnosis was 42 years [38]. The youngest patients were *RET* pathogenic variants carriers who underwent (genetic) screening with median ages of 12 and 14 years [39]. The pattern of inheritance was shown to be of importance in MEN2-PHPT, as follows: a higher risk of developing PHPT in *RET* carriers was found in those who inherited the condition from the father versus those who inherited it from the mother, according to a hazard ratio (HR) of 3.4 [95% confidence interval (CI) between 1.1 and 10.1, *p* = 0.029] [44].

### 3.1. Genetic Findings: RET-Related Primary Hyperparathyroidism

Genetic testing plays a crucial role in familial (hereditary) PHPT, as seen in other endocrine (non-parathyroid) conditions [49,50,51]. Ten (n = 10/14) studies [35,36,38,39,40,41,43,45,47,48] provided the *RET* pathogenic variants involved in MEN2 or the risk category, according to American Thyroid Association (ATA) criteria [49]. Genetic analysis was reported in a total of 3399 subjects, including 3163 with MEN2, out of whom 177 individuals had MEN2-related PHPT. Holm et al. [38] provided the prevalence of symptoms according to the *RET* pathogenic variant and found that 16.67% of patients with p.Cys634Arg and 37.5% of those with p.Cys611Tyr had symptomatic PHPT, while those with p.Cys618Phe and p.Leu790Phe were asymptomatic [38]. A retrospective study showed that index cases with MEN2-asociated PHPT as first manifestation of the syndrome had a symptomatic PHPT form in relationship with the following pathogenic variants: p.Cys634Tyr, p.Cys634Arg, p.Cys611Tyr, p.Cys620Arg, p.Glu768Asp, and p.Cys618Phe [47].

In terms of the number of glands involved, Holm et al. [38] reported single-glandular disease in patients harboring the *RET* mutation at p.Cys634Arg, p.Cys611Tyr, and p.Cys618Phe, and multi-glandular disease in relationship with p.Cys634Arg, p.Cys611Tyr, and p.Leu790Phe [38], while Larsen et al. [47] found multi-gland involvement in subjects with PHPT as first MEN2 manifestation with respect to p.Cys634Tyr and p.Cys634Arg pathogenic variants, and single-gland disease in p.Cys634Tyr, p.Cys634Arg, p.Cys611Tyr, p.Cys620Arg, p.Glu768Asp, and p.Cys618 [47].

Some studies assessed PHPT with regard to the *RET* pathogenic variant-associated risk category for MTC [39,40]. High-risk variants had a higher prevalence of PHPT compared with moderate-high risk variants and low-risk variants (11.4% versus 2.4% versus 0.5%, *p* < 0.001). However, the age at PTx did not have a statistically significant difference (*p* = 0.270) among these mentioned subgroups of analysis. In high-risk variants (for MTC), PHPT prevalence was lower during the recent years (*p* < 0.001) [40]. Moreover, the age at PTx in index cases versus non-index cases was similar, as similarly found in individuals with high-risk pathogenic variants (*p* = 0.370) or moderate-high risk (*p* = 0.980) for MTC [39]. Another cohort from 2023 found a statistically significant increased prevalence of PHPT in patients harboring p.Cys634Arg/Thr/Tyr versus p.Cys618Arg (33.3% versus 3.2%, *p* = 0.01) [41] (Table 2).

### 3.2. The Clinical Presentation and Spectrum of Complications in MEN2-Related Primary Hyperparathyroidism

Data regarding symptoms/clinical presentation of PHPT in MEN2 subjects were provided by 5/14 studies (N = 49 patients with MEN2-related PHPT) [36,37,38,46,47]. The most frequent clinical finding/complication was nephrolithiasis with a prevalence of up to 80% [47], followed by osteoporosis with the maximum prevalence of 12.5% [38]. Other symptoms included pancreatitis [46], polyuria [47], and non-specific clinical picture [37], while chronic kidney disease was reported in 9.1% of the individuals in one cohort [36]. Although most studies reported symptomatic PHPT, Holm et al. [38] found that 75% of patients with PHPT were asymptomatic [38]. Of note, apart from PHPT-related symptoms, Larsen et al. [47] also reported the timing of MTC diagnosis, which was synchronous with PHPT diagnosis in 80% and metachronous in 10% of the patients [47] (Table 3).

### 3.3. Parathyroidectomy in MEN2 Subjects

Ten studies (n = 10/14) reported data regarding PTx (N = 156 subjects with MEN2-related PHPT) [36,37,38,39,40,42,44,46,47,48]. Between 72.7% [36] and 100% [37,39,42,47] of PHPT patients underwent PTx as curative treatment for the underlying parathyroid tumors. PTx was typically performed in adulthood at ages spanning from a mean of 34.7 to 48.5 years [39]. Based on the specific *RET* pathogenic variant, there were no statistically significant differences regarding the age at PTx (*p* = 0.270) [40]. When index and non-index patients were compared, the age at PTx was similar, as well [39]. The parent who transmitted the gene, however, influenced the age at PTx, as was shown by Machens et al. [44] who revealed that subjects who inherited the pathogenic variant from the father had younger age at PTx compared to those who inherited from the mother (P_log-rank_ = 0.018) [44].

Furthermore, in recent years, PTx was performed at younger ages compared to the past as shown by a retrospective study (e.g., the age at PTx decreased from 43.5 years in ≤1950 to 16.5 years in the 1991–2000 birth cohorts with high-risk pathogenic variants) [40], while a cross-sectional study confirmed the same outcome [age at PTx, median (IQR) by birth cohort: 1922–1950 versus 1951–1960 versus 1961–1970 versus 1971–1980 versus 1981–1990 versus 1991–2000 versus 2001–2010: 46 (39.5–55) versus 42 (31–45.5) versus 31 (23–36) versus 26 (26–26) versus 12 (12–12) y, *p* = 0.008] [48].

With respect to the surgical technique (n = 4/14 studies, N = 32 patients who underwent PTx), selective PTx was performed most commonly [36]. Subtotal PTx was the second most used technique, with the highest prevalence of 69% in a population-based study by Holm et al. [38]. Other procedures included subtotal PTx with auto-transplantation of the parathyroid tissue in the forearm [38] and bilateral neck exploration [37].

The outcomes of PTx varied (only three studies reported the postoperative outcomes), for example, the post-surgery rates of persistent PHPT was of 0% [36], 8% [38], and 16.7% [48]; recurrent PHPT was of 12.5% [36] and 23% [38]; permanent hypoparathyroidism was reported in 46% [38] and 33.3%, respectively [44]; permanent unilateral vocal cord palsy was not identified in one study [38], but another reported it in 16.7% of the subjects [44] (Table 4).

### 3.4. Histological Analysis of the Parathyroid Tumors

Data regarding the number of glands affected were reported by seven studies (N = 77 patients who underwent PTx) [36,37,38,39,42,46,47]. The prevalence of multi-glandular disease ranged between 12.5% [36] and 50% [42,46], while the rate of uni-glandular disease was of 33.3% [42] and 87.5% [36], respectively. In terms of parathyroid tumor size, two studies provided these specific data [37,39]: Gasior et al. [37] found a median (IQR) tumor size of 0.7 (0.55–0.9) cm, with a median (IQR) tumor mass of 118 (56.3–302) mg [37], while Machens et al. [39] reported similar tumor diameters in index and non-index cases of 3.72 (2.98–4.47) and 4.07 (3.39–4.75) cm, respectively (*p* = 0.505) [39] (Table 5).

### 3.5. Imaging Assessment in Patients with MEN2-Associated Primary Hyperparathyroidism

Three studies [37,42,46] reported imaging findings from 16 patients with PHPT due to MEN2, including preoperative localization [37,46] and intraoperative imaging [42]. Gasior et al. [37] reported that preoperative localization as follows: 40% of patients had ultrasound assessment, and 40% underwent Tc-98m Sestamibi scans [37]. However, preoperative localization was not always successful as it was shown by Diwaker et al. [46] who reported a successful rate in 29% of patients [46]. Berber et al. [46] investigated auto-fluorescence signals during PTx, a more recent technique in PHPT, including MEN2 patients, and reported a median auto-fluorescence intensity of 1.8 and a median heterogeneity index of 0.11 [46].

### 3.6. Primary Hyperparathyroidism in MEN2 Versus Other Familial Syndromes

Figueiredo et al. [36] investigated differences among different familial forms of PHPT in a retrospective analysis on 48 subjects with familial PHPT, including 11 individuals with PHPT in the setting of MEN2. When PHPT in MEN2 was compared with MEN1, there was no statistically significant difference in the prevalence of PHPT as first manifestation of the syndrome (*p* = 0.13). However, serum parathormone (PTH) was lower (median of 108.0 versus 196.9 pg/mL, *p* = 0.01), serum calcium levels were lower (mean ± SD: 10.6 ± 1.1 versus 11.7 ± 1.2 mg/dL, *p* = 0.03), and less parathyroid glands were affected [median ± standard deviation (SD): 1.1 ± 0.3 versus 2.7 ± 0.9, *p* < 0.001] in MEN2 compared to MEN1 [36].

In MEN2, PHPT was the first manifestation less frequently compared to hyperparathyroidism-jaw tumor syndrome (0% versus 85%, *p* = 0.001), while serum PTH (median: 108.0 versus 383.5 pg/mL, *p* = 0.01) and serum total calcium (mean ± SD: 10.6 ± 1.1 versus 12.9 ± 1.8 mg/dL, *p* < 0.001) levels were lower, and nephrolithiasis occurred less often (18.2% versus 65%, *p* = 0.02). The number of parathyroid glands was similar (1.1 ± 0.3 versus 1.6 ± 1.1, *p* = 0.23) [36].

### 3.7. Medullary Thyroid Carcinoma in MEN2 Patients (The Data According to the Studies That Also Provided an Analysis of the Primary Hyperparathyroidism)

Ten studies (n = 10/14; N = 3760 patients) analyzed the features of MTC amid MEN2 confirmation [35,38,39,40,41,43,44,46,47,48]; the highest prevalence of MTC reached 100% [47] and varied according to the *RET* pathogenic variant, with maximum rate in p.Met918Thr (100%) and p.Cys634Phe/Gly/Arg/Ser/Trp/Tyr (88.9%) [35]. Machens et al. [40] reported a higher prevalence in high-risk pathogenic variants (75% versus 65.2% versus 63.2%, *p* = 0.016) and younger age at thyroidectomy (17 versus 29 versus 39 years, *p* < 0.001) [40]. A multicenter study showed that *RET* pathogenic variants correlated with the tumor size; variants affecting codon C634 being associated with larger tumors compared with those involving codon C618 (1.85 ± 1.11 versus 0.89 ± 0.67 cm, *p* = 0.004) and with higher calcitonin levels as well (333.9 ± 314.5 versus 84.5 ± 201.9 ng/mL, *p* = 0.030) [41]. The prevalence of C-cell hyperplasia versus MTC confirmation was reported by Holm et al. [38] at 31% versus 69% [38]; similar findings were also revealed by a study on p.Cys634 carriers (76.5% versus 21.6%) [48].

MEN2 individuals who were diagnosed through screening protocols were younger compared with those with hereditary MTC (28.3 ± 19.8 versus 30.15 ± 15.3 years, *p* < 0.05); they also had smaller tumors compared with sporadic MTC and MEN2 index cases (2.9 ± 0.85 versus 3.14 ± 1.43 versus 2.96 ± 1.38 cm) [46]. Index cases had a higher prevalence of MTC (97.4% versus 57.0%, *p* < 0.001), as well as larger tumors (1.95 versus 0.79 cm, *p* < 0.001], higher rates of lymph nodes metastases (71.5% versus 29.5%, *p* < 0.001), and lower rates of biochemical cure (34.1% versus 74.8%, *p* < 0.001) [39]. Similar MTC prevalence were highlighted in subjects who inherited the *RET* variant from the mother or the father [43,44], while lymph node metastases were more frequent in patients who received the gene pathogenic variant from the father compared with the mother (45% versus 19%, *p* = 0.006 and 43% versus 29%, *p* = 0.029) [43,44] (Table 6).

### 3.8. MEN2-Associated Pheochromocytoma (The Data According to the Studies That Also Provided an Analysis of the Primary Hyperparathyroidism)

MEN2-related PC analysis was provided by eleven studies (n = 11/14, N = 4026 patients) [35,38,39,40,41,43,44,45,46,47,48]. High-risk *RET* pathogenic variants had the highest prevalence of PC, of 55.6% (in p.Cys634Phe/Gly/Arg/Ser/Trp/Tyr) and of 50% (in p.Met918Thr) [35]. These findings are supported by another study that reported a higher prevalence of PC in high-risk pathogenic variants compared with moderate-high and low-risk groups (32.1% versus 16.4% versus 3%, *p* < 0.001) [40]. Moreover, p.Cys634Arg/Thr/Tyr also had a higher prevalence of PC compared with p.Cys618Arg (53.3% versus 6.5%, *p* = 0.001) [41]. Milcevic et al. [45] found that, while *RET* pathogenic variants such as p.Cys634Phe/Gly/Arg/Ser/Trp/Tyr and p.Cys618Phe/Arg/Ser had a PC prevalence of 70.6% and 4.5%, respectively, PC did not occur in other variants such as p.Leu790Phe, p.Val804Met, p.Ser891Ala, and p.Met918Thr [45].

Synchronous diagnosis of PHPT and PC was found in 56%, while PC was identified before PHPT in 22% of the cases, respectively; PC was confirmed after PHPT in 22% [38]. Index cases had a higher prevalence of PC compared to non-index cases (30.1% versus 13.2%, *p* < 0.001), but similar ages at adrenalectomy (*p* = 0.431) [39]. PC had the same prevalence in subjects who inherited the variant from the mother compared with the father origin (19% versus 33%, *p* = 0.051 [43] and 13% versus 19%, *p* = 0.094 [44]). However, there was a higher prevalence of bilateral PC in those who inherited the pathogenic variant from the father (24% versus 10%, *p* = 0.021) [43]). Other MEN2-related PC findings included a prevalence of 48.64% [46] and a PC rate in the second adrenal gland in 18.8% of MEN2 patients [48] (Table 7).

## 4. Discussion

### 4.1. Inherited Forms of Primary Hyperparathyroidism

Syndromic (genetic or hereditary) combinations of endocrine ailments, either involving autoimmune or tumor/cancer features, still represents a multidisciplinary challenge nowadays [52,53,54,55]. A total of 5–10% of PHPT patients may have a familial (monogenic) form of disease [56]. Hereditary PHPT occurs either as the sole endocrine condition (as found in familial isolated PHPT), or as a syndromic type. Genetic syndromes that promote the development of PHPT have an autosomal dominant inheritance pattern and include MEN1, MEN2, MEN4, and hyperparathyroidism-jaw tumor syndrome [57,58,59]. While MEN1, MEN4, and hyperparathyroidism-jaw tumor syndrome involve inactivating pathogenic variants of tumor suppressor genes *MEN1*, *CDKN1B*, and *CDC73,* respectively, MEN2 is caused by activating mutations of *RET* proto-oncogene [60,61,62,63]. These syndromes often present a variety of symptom clusters and multi-layered complications; hence, their management represents a complex process [64,65,66]. Differentiating between the familial causes of PHPT is one of the most important steps for an adequate multimodal management and active search/screening for other associated endocrine and non-endocrine manifestations of the syndromes, as well as for initiating a genetic screening among the family members [66].

While PHPT is a cardinal feature of hyperparathyroidism-jaw tumor syndrome [67] and it is the main manifestation of MEN1, affecting over 90% of patients [68], the prevalence of PHPT in MEN2 is lower, with a penetrance around 30% (e.g., recent data reported a prevalence of up to 31.3% [45], as identified across our analysis). Another key difference between these syndromes is the number of parathyroid glands involved; typically, familial syndromes have multi-glandular involvement [69,70]. However, our analysis highlighted unexpected high rates of single-gland disease, as mentioned by some studies [36,37,38,47]. Despite a low frequency for a multi-gland involvement, PHPT post-surgery recurrence was increased, up to 23% [38].

In MEN2 subjects, PHPT is rarely the first manifested endocrine disorder [47]. Recognizing MEN2 in these patients is extremely important, so that MTC screening may be early performed for a better overall prognosis [71]. Figueiredo et al. [36] highlighted key differences between distinct hereditary PHPT forms and found that the lowest prevalence was among familial PHPT (of 22.9%) compared with 41.7% in patients with hyperparathyroidism-jaw tumor syndrome, or 35.4% in MEN1 [36]. The importance of PHPT diagnosis in MEN2 patients should not be downplayed by the relatively less severe presentation compared with other syndromic manifestations such as the presence of a potentially severe thyroid malignancy, especially if PHPT co-occurs with MTC, making preoperative diagnosis of both conditions crucial for the surgical approach [72,73]. Notably, a case–control analysis of the serum calcium levels in PHPT amid MEN2 confirmation (in order to highlight the impact of acute hypercalcemia and a potentially more severe presentation) was not provided by all the studies we could identify [35,38,39,40,41,42,43] or the results only included data with a relatively low statistical power [37]. On the other hand, as mentioned, Figueiredo et al. [36] showed that serum total calcium and PTH were statistically significantly lower in MEN2 than MEN1, of 10.6 10.6 ± 1.1 versus 11.7 ± 1.2 mg/dL (*p* = 0.03) and 108 versus 196.9 pg/mL (*p* = 0.01), respectively, but, also, than found in hyperparathyroidism-jaw tumor syndrome, of 12.9 ± 1.8 mg/dL (*p* < 0.001) and 383.5 pg/mL (*p* = 0.01) [36], respectively.

#### Primary Hyperparathyroidism in the Setting of MEN2

MEN2 leads to the development of neoplasia in the thyroid, adrenals, and parathyroid glands. The first clinical manifestation, with the highest penetrance, occurring in virtually all patients with MEN2, is MTC [74,75,76]. The strong link between *RET* and MEN2 is further highlighted by the prevalence of germline *RET* pathogenic variants in MTC patients in the general population, which exceeds 16% [77]. Considering that most patients with MEN2 are typically diagnosed with MTC during childhood, many undergo prophylactic thyroidectomy at an early age [78,79,80]. In order to achieve the best cure rate, *RET* genetic screening and early diagnosis, as well as identifying the pathogenic variant risk category for better surgical planning, are essential [80].

According to ATA guidelines, the risk of MTC is stratified by the *RET* pathogenic variant in distinct categories, the highest being in codon M918T, while codon C634 and codon A883F also involve a high risk of MTC [78,79,80]. Recent data analyzed characteristics of PHPT based on this classification. As mentioned, two studies reported a higher prevalence of PHPT in high-risk pathogenic variants [40,41]. However, the age at PTx was similar across these pathogenic variant groups [40].

A new classification of *RET* variants associated with MEN2 has been recently proposed by the American College of Medical Genetics and Genomics and the Association for Molecular Pathology and includes the introduction of two new categories, likely benign and likely pathogenic, as well as the re-classification of the significance in certain *RET* variants [81].

In association with MTC and PHPT, PC affects 30% of the MEN2 subjects; its penetrance is connected to the MTC aggressiveness [82]. PC usually manifests during the third decade of life, and due to the life-threatening complications, early detection across the biochemical screening is advised [83,84]. Identifying PC prior to other surgical interventions is crucial, as hidden/unrecognized PC may receive inappropriate alpha-adrenergic receptor blockade, with severe perioperative consequences, including cardiac arrest and fatal outcome [85]. Surgical treatment requires preoperative blood pressure control and the prevention of intraoperative hypertensive crises, and the presence of PHPT-related hypercalcemia should be carefully taken into consideration as well [86]. When synchronously occurring with MTC or PHPT, PC needs immediate attention and adrenalectomy should be performed before thyroidectomy or PTx [87]. In some cases, simultaneous (single-time) adrenalectomy and thyroidectomy have successfully been performed, and it requires an experienced surgical team [88].

PHPT, a less frequent manifestation of MEN2, associates, however, a much higher prevalence than seen in the general population which, although rising, is less than 1% [89,90]. Typically, sporadic PHPT affects menopausal women, while hereditary PHPT manifests at much younger ages with similar prevalence in men and women [2,91]. In MEN2, however, while the prevalence in men and women is similar, PHPT usually manifests later in life compared to other familial syndromes, in the third and fourth decades of life, as also reflected by our sample-based analysis [35,36,38,40,47,48].

Typical symptoms of PHPT include nephrolithiasis, low-trauma/spontaneous fractures and osteoporosis, cardio-metabolic complications, non-specific symptoms such as digestive symptoms, and fatigue, as well as various clinical features that are induced by acute hypercalcemia [92,93,94]. Yet, nowadays, due to calcium screening protocols, the most often presentation is asymptomatic or mildly symptomatic PHPT [95,96,97]. Another disease form is normocalcemic PHPT, often diagnosed during supplementary investigations of subjects with kidney stones, low bone mineral density, or incidental fractures, an entity characterized by normal levels of albumin-adjusted serum calcium and ionized calcium and high levels of PTH in the absence of secondary causes of high PTH [96,98,99,100,101]. Recent data (across our methods of search) did not particularly report or explore this disease form in MEN2-PHPT, and future studies are needed. As mentioned, classical symptoms of PHPT such as nephrolithiasis and osteoporosis were described in some studies [38,47]. Additionally, about 6% of patients with PHPT may develop pancreatitis [46], a manifestation that might be the sole indication of PHPT in some cases [102,103]. Moreover, non-classical features in PHPT (e.g., depression, anxiety, cardiovascular manifestations, or glucose metabolism changes [104,105]) were not distinctly analyzed in this MEN2-PHPT-focused analysis.

Of note, PHPT increases the fragility fracture risk due to cortical bone loss, especially of the distal radius, as well as altered bone microarchitecture, as reflected by low trabecular bone score [106,107,108]. Therefore, the presence of osteoporosis represents a surgical indication in PHPT [109]. Holm et al. [38] identified that 12.5% of the PHPT subjects had osteoporosis [38]; however, we could not identify any more specific data across the other studies regarding the bone status.

### 4.2. Surgery Candidates

Surgical indications in MEN2-related PHPT are similar to those applied to PHPT in the general population (e.g., the identification of skeletal or renal complications, serum total calcium levels higher than 1 mg/dL above the upper limit or patient age below 50 years, etc.) [109]. Surgical planning is highly individualized and has several particularities. For instance, PTx often needs to be performed on an already explored neck, making the surgery more difficult and exposing the patient to a higher rate of complications [110,111]. Typically, the first manifestation of MEN2 is MTC, a severe disease with a high mortality and morbidity that requires total thyroidectomy with prophylactic or curative lateral neck dissection [112,113,114]. Due to the high penetrance of MTC, especially in high-risk variants, prophylactic thyroidectomy is practiced in *RET* pathogenic variants carriers at very early ages [115,116]. While prophylactic thyroidectomy in children may be performed without lymph node dissection in certain stages/mutations, MTC surgery may need large neck dissection with multiple postoperative complications [117,118,119]. Hence, a future lateral neck exploration is more difficult due to fibrosis, etc. [120,121].

Notably, preoperative imaging scans represent another crucial step for surgical planning, especially considering the multi-glandular disease in PHPT (with asynchronous presentation) [47,122]. Moreover, the preoperative assessment should include the evaluation of previous forearm grafts from prior PTxs, and of possible complications from prior surgery(s) such as laryngeal nerve palsy, etc. [122].

Prophylactic PTx in MEN2 patients who undergo thyroidectomy for MTC was explored in the past, but it is not currently supported due to the fact that PHPT is actually diagnosed later in life (during adulthood) and PHPT penetrance is only 30% [123]. Older data showed that, while some authors are in support of total PTx and auto-transplantation since it seems a safe approach [124,125], nowadays, if parathyroid glands are macroscopically normal, we consider that they should not be removed, noting the low rate of recurrence and of surgical complications [126,127,128,129,130]. Currently, a less aggressive and minimally invasive approach is preferred in order to reduce the post-operatory rate of complications, including hypocalcemia and hypoparathyroidism [127,128,129,130]. The use of intra-operatory PTH assays helps the rate of complete gland/tumor removal and avoids unnecessary redo surgery [126,127,128,129,130].

If total PTx with auto-transplantation is performed, regular follow-up is still needed, due to PHPT recurrence or multi-gland involvement, including in ectopic parathyroid [129,130]. As mentioned, we identified a single study that explored intraoperative auto-fluorescence [42], a novel technique used for identifying parathyroid glands based on the natural fluorescence emitted by the parathyroid tissue during exposure to near-infrared light (which is especially useful in bilateral neck exploration) [131,132,133,134]. The clinical and therapeutic implications in MEN2 patients, however, remain an emerging topic that needs further studies. No additional data with respect to using intraoperative indocyanine green angiography for glands localization in MEN2 we could identify, neither in using cryopreservation, but these seem promising alternatives for selected cases amid a tailored multimodal management in MEN2-related PHPT.

### 4.3. Case Report-Focused Analysis

Exploring the fascinating domain of PTH, from PTH-producing tumors amid genetic conditions to the practical use of PTH analogues in daily endocrine practice [135], might pinpoint aspects with a lower level of statistical evidence. Hence, across our search, we identified a collateral result in terms of five novel reports [136,137,138,139,140] in MEN2-PHPT in subjects who underwent genetic testing for *RET* pathogenic variants. Among them, four cases included adult patients (three males and one female) with ages between 26 and 68 years, diagnosed with PHPT in the setting of MEN2 [136,137,138,139]. All patients were surgically treated for PHPT: selective PTx was performed in one patient [136], while another underwent total PTx with re-implantation in the forearm [139]. Thoracoscopy was necessary in the case of a 26-year-old male for an ectopic parathyroid mass, after having undergone a prior PTx with post-surgery persistent PHPT. Following the resection of the ectopic mass, the patient developed hypoparathyroidism [137]. Synchronous MTC occurred in 3/5 subjects requiring thyroidectomy [136.137,139]. Two patients associated PC [136,137], while another had a pancreatic paraganglioma [138]. Family history was positive only in 1/5 subjects [139]. However, while the other three did not have a family history of endocrine neoplasia [136,137,138], another had three relatives positive for the same *RET* pathogenic variant [136].

The complexity of hereditary forms of PHPT was reflected by a 28-year-old male who developed asymptomatic PHPT, multifocal MTC, and bilateral PC in the setting of two pathogenic variants of the *RET* gene (Cys630Tyr) and of the *MEN1* gene (p.Ala176Leufs*10) [139]. One case in particular highlighted the phenotypic variability and interconnection between MEN2 and MEN3 manifestations. In this case of a 7-year-old girl, reported by Giani et al. [140], the p.Asp631_Leu633delinsGlu de novo *RET* variant manifested suggestive features for MEN2 (e.g., PHPT) and distinct aspects of MEN3 (including marfanoid habitus and mucosal neuromas). In addition, the patient had MTC and a history of plexiform neurofibroma and ganglioneuromatosis [140] (Table 8).

### 4.4. Limits and Further Expansion

This sample-focused analysis was introduced across a non-systematic review in order to not restrain the original studies data to similar statistical parameters, a type of analysis which is less likely feasible at this point in the field of MEN2-PHPT. As mentioned across this narrative review, our search did not detect distinct results in terms of PHPT-connected clinical elements such as osteoporosis or fracture prevalence, etc. Further longitudinal studies are necessary to assess the long-term outcome in patients with MEN2-PHPT. Moreover, we raise the issue of epidemiologic data with concern to new PHPT subtypes such as normocalcemic or asymptomatic in MEN2 individuals. As specified, novel surgical procedures, including minimally invasive approaches, might minimize the impact of the surgery in these patients and provide a better outcome (Figure 2).

## 5. Conclusions

PHPT in MEN2 patients may occasionally present as the first manifestation of the syndrome. In contrast to other familial forms of PHPT, it usually manifests later in life during the third and fourth decades and frequently has single-gland involvement. In spite of a relatively low prevalence compared to the other manifestations of MEN2, PHPT should not be overlooked considering the chance of recurrence and the high frequency in high-risk mutations. Surgical planning needs to be tailored to every case and takes into consideration previous neck surgery, possible preexisting complications, and the transformation of transplanted parathyroid tissue. Apart from early diagnosis and surgical treatment, life-long follow-up represents the key to the management of PHPT in MEN2 individuals.

## Figures and Tables

**Figure 1 diseases-13-00098-f001:**
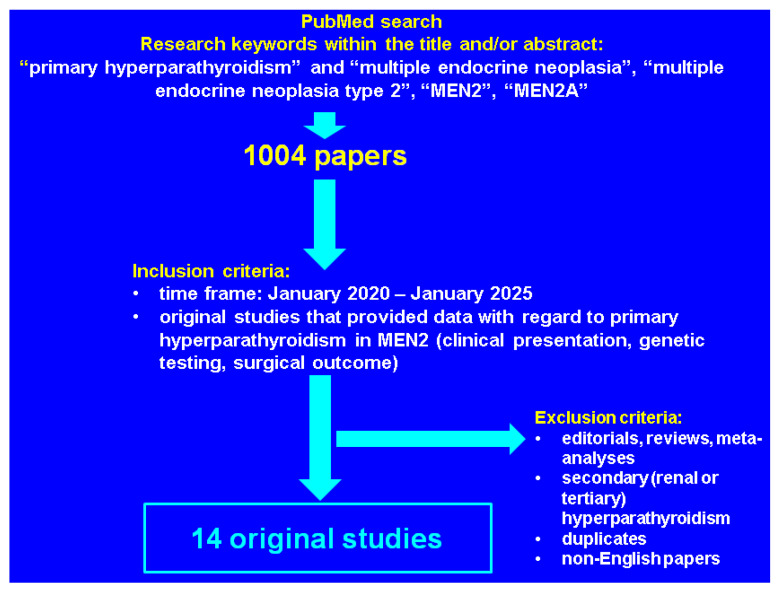
Flowchart of literature research according to our methods.

**Figure 2 diseases-13-00098-f002:**
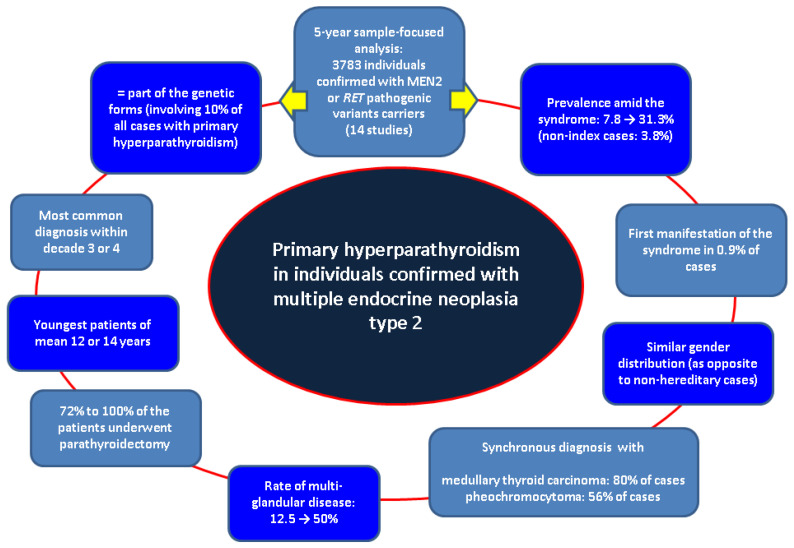
Flowchart of the main findings across our methods [35,36,37,38,39,40,41,42,43,44,45,46,47,48].

**Table 1 diseases-13-00098-t001:** Included studies with concern to MEN2-related PHPT [35,36,37,38,39,40,41,42,43,44,45,46,47,48] (the display starts with the most recent publication year).

First Author Year of Publication Reference Number	Study Design and Studied Population	Prevalence of PHPT Among MEN2 Patients
Binter 2024 [35]	Retrospective cross-sectional single-center study N = 158 with MEN2 Median (min, max) age = 53 (3–91) y N1 = 13 with PHPT Median (min, max) age at PTx = 33 (14, 72)	8.22% (13/158)
Figueiredo 2023 [36]	Retrospective analysis N = 48 with familial form of PHPT Mean ± SD age = 40 ± 15.5 y F:M = 24:24 (50% females) N1 = 11 (22.9%) with PHPT due to MEN2 Age at PHPT diagnosis (mean ± SD) = 43.9 ± 19.2 y Age at first manifestation (mean ± SD) = 31.3 ± 19.5 y F:M = 6:5 (54.5% females)	NA
Gasior 2023 [37]	Retrospective study N = 3889 with PHPT who underwent PTx Age, mean ± SD = 59 ± 13 y F:M = 3000:889 (77.1% females) N1 = 5 (0.12%) with MEN2 Age, mean ± SD = 33.4 ± 10.6 y F:M = 4:1 (80% females)	NA
Holm 2023 [38]	Population-based retrospective study N = 204 with MEN2 N1 = 16 with PHPT due to MEN2 Age at MEN2 diagnosis, median (IQR) = 42 (4–82) y Age at PHPT diagnosis, median (IQR) = 45 (21–79) y F:M = 5:11 (31.25% females)	7.84% (16/204)
Machens 2023 [39]	Cross-sectional study N = 602 MEN2 carriers F:M = 341:261 (56.64% females) N1 = 156 index cases N2 = 446 non-index-patients Mean (95% CI) age at PTx: N1 vs. N2: 40.9. (33.7–48.0) vs. 36.4 (29.3–43.4), *p* = 0.348 N3 = 31 with MEN2 and PHPT (14 N1 and 17 N2)	N1 vs. N2: 9% (14/156) vs. 3.8% (17/446), *p* = 0.019
Machens 2023 [40]	Retrospective observational study N = 604 with MEN2 who underwent surgery for MEN2 High-risk (N1 = 237) vs. moderate-high risk (N2 = 165) vs. low risk (N3 = 202) pathogenic variants: Age at most recent follow-up, median (IQR): 26 (10.5–40) vs. 30 (11.5–49.5) vs. 43 y, *p* < 0.001 N4 = 32 with PHPT and MEN2 N4 out of N1 vs. N2 vs. N3: Age at PTx, median (IQR): 37 (25–46) vs. 48.5 (35.3–62.5) vs. 43 y, *p* = 0.270 F:M = 130:107 (54.9% females) vs. 97:68 (58.6% females) vs. 114:86 (57.4% females), *p* = 0.720	Prevalence of PHPT, according to pathogenic variant risk category: High risk: 11.4% (27/235) Moderate-high risk: 2.4% (4/165) Low risk: 0.5% (1/202) *p* < 0.001 PHPT by pathogenic variants by year: ≤1950 vs. 1951–1960 vs. 1961–1970 vs. 1971–1980 vs. 1981–1990 vs. 1991–2000 vs. 2001–2010 vs. 2011–2020: High risk: 42% vs. 20% vs. 23% vs. 3% vs. 3% vs. 6% vs. 0% vs. 0%, *p* < 0.001 Moderate-high risk: 3% vs. 9% vs. 5% vs. 0% vs. 0% vs. 0% vs. 0% vs. 0%, *p* = 0.540 Low risk: 0% vs. 0% vs. 0% vs. 4% vs. 0% vs. 0% vs. 0% vs. 0%
Rosenblum 2023 [41]	Retrospective multicenter study N = 50 with MEN2 and MTC N1 = 31 with p.Cys618Arg N2 = 15 with p.Cys634Arg/Thr/Tyr Age at MTC surgery, mean ± SD = 25.7 ± 10.9 vs. 31.3 ± 17.5 y, *p* = 0.190 N3 = 6 with PHPT	N: 8.33% (6/50) N1: 3.2% (1/31) N2: 33.3% (5/15) N1 vs. N2: *p* = 0.010
Berber 2022 [42]	Retrospective analysis N = 183 with PHPT who underwent PTx N2 = 4 (48%) with PHPT due to MEN2	NA
Machens 2022 [43]	Retrospective analysis N = 169 carriers of *RET* missesense pathogenic variants N1 = 90 with affected mother F:M = 53:37 (59% females) Age at most recent follow-up, median (IQR) = 14 (7–34) y N2 = 79 with affected father F:M = 45:34 (57% females) Age at most recent follow-up, median (IQR) = 12 (6–25) y N3 = 14 with PHPT (4 in N1, 10 in N2) F:M = 6:8 (42.85% females)	N1 vs. N2: 4% (4/90) vs. 13% (10/59), *p* = 0.090 F vs. M: N1: 6% (3/53) vs. 3% (1/37), *p* = 0.641 N2: 7% (3/45) vs. 21% (7/34), *p* = 0.090
Machens 2022 [44]	Observational study N = 578 *RET* carriers who underwent surgery for MEN2 related tumors N1 = 236 with affected mother F:M = 141:95 (60% females) Age at most recent follow-up, median (IQR): 23 (10, 36) y N2 = 169 with affected father F:M = 94:75 (56% females) Age at most recent follow-up, median (IQR): 24 (10, 38) y N3 = 29 carriers with PHPT (6 in N1, 11 in N2)	5% (29/578) N1 vs. N2: 3% (6/236) vs. 7% (11/169), *p* = 0.076Risk of PHPT: Offspring sex M vs. F: HR (95% CI) = 1.0 (0.4–2.6), *p* = 0.960 Index vs. non-index-patients: HR (95% CI) = 1.2 (0.4–3.6), *p* = 0.758 Parental inheritance father vs. mother: HR (95% CI) = 3.4 (1.1–10.1), *p* = 0.029
Milicevic 2022 [45]	Retrospective analysis N = 266 with MTC and relatives N1 = 208 with MTC 21.6% (45/208) *RET* positive, 64.4% (134/208) *RET* negative, 4.8% (10/208) with *RET* variant of unknown significance, and 9.1% (19/208) without genetic testing F:M = 110:98 (52.9% females) Mean ±SD age at diagnosis in positive vs. *RET* negative: 42 ± 19 y vs. 59.8 ± 13.9 y, *p* < 0.001 N2 = 67 positive individuals, pertaining to 21 families, out of 103 tested individuals (MTC patients and family members) N3 = 21 with PHPT and MEN2	31.34% (21/67)
Diwaker 2021 [46]	Retrospective study N = 97 with MTC N1 = 46 with hereditary MTC [F:M = 22:24 (48% females) Age at MTC diagnosis, mean ± SD = 30.15 ± 15.3 y] N2 = 51 with sporadic MTC [F:M = 30:21 (59% females) Age at MTC diagnosis, mean ± SD = 40.09 ± 14.78 y] N3 = 37 with MTC due to MEN2 [F:M = 20:17 (54% females)] N4 = 14 index cases with MEN2 [F:M = 11:14 (44% females) Age at MTC diagnosis, mean ± SD = 35.3 ± 11.9 y] N5 = 12 diagnosed by familial screening [F:M = 9:3 (75% females) Age at MTC diagnosis, mean ± SD = 28.3 ± 19.8 y] Age at MTC diagnosis: N2 vs. N1 *p* < 0.05, N2 vs. N4 *p* < 0.05, and N2 vs. N5 *p* < 0.05 N6 = 7 with PHPT due to MEN2	In N3: 19% (7/37), with F:M = 4:3
Larsen 2020 [47]	International multicenter retrospective study N = 1085 index cases with MEN2 N1 = 10 with PHPT as first manifestation F:M = 8:2 (80% females) Age at PHPT diagnosis, median (IQR) = 34.5 (14–68) y	PHPT as first manifestation: 0.9% (10/1085)
Machens 2020 [48]	Cross-sectional study N = 213 *RET* pathogenic variant p.Cys634 carriers F:M = 118:95 (55.4% females) Age at last follow-up, median (IQR) = 26 (11–40) y N1 = 23 with PHPT Age at PTx, median (IQR) = 39 (26–46) y	10.8% (23/213) 1922–1950 vs. 1951–1960 vs. 1961–1970 vs. 1971–1980 vs. 1981–1990 vs. 1991–2000 vs. 2001–2010 PHPT: 41% vs. 22% vs. 21% vs. 0% vs. 3% vs. 3% vs. 0%, *p* < 0.001

Abbreviations: CI = confidence interval; F = female; HR = hazard ratio; IQR = interquartile range; M = male; MEN2 = multiple endocrine neoplasia type 2; MTC = medullary thyroid carcinoma; N = number of patients; NA = not available; PHPT = primary hyperparathyroidism; PTx = parathyroidectomy; SD = standard deviation; vs. = versus; y = years (red font and blue font represent different study subgroups of analysis).

**Table 2 diseases-13-00098-t002:** *RET* pathogenic variants in patients with primary hyperparathyroidism amid MEN2 confirmation [35,36,38,39,40,41,43,45,47,48].

Reference	Main Findings *
[35]	 **Prevalence of PHPT, according to pathogenic variant:**p.Cys634Phe/Gly/Arg/Ser/Trp/Tyr: 25.9% (7/27)p.Cys611Phe/Gly/Arg/Ser/Tyr: 4% (1/25)p.Cys630Arg/Tyr: 100% (1/1)p.Val804Met/Leu: 7.1% (4/56)  **Median (min, max) age at PHPT diagnosis, according to pathogenic variant:**p.Cys634Phe/Gly/Arg/Ser/Trp/Tyr: 33 (14–72) yp.Cys611Phe/Gly/Arg/Ser/Tyr: 23 yp.Cys630Arg/Tyr: 23 yp.Val804Met/Leu: 59.5 (30–79) y
[36]	 **Number of PHPT patients according to pathogenic variant:**p.Cys611Thr: 2p.Cys634Arg: 4p.Cys634Thr: 2
[38]	 **Prevalence of different mutations in N1:**p.Cys634Arg: 37.5% (6/16)p.Cys611Tyr: 50% (8/16)p.Cys618Phe: 6.25% (1/16)p.Leu790Phe: 6.25% (1/16)  **Prevalence of symptoms:**p.Cys634Arg: 16.67% (1/6)p.Cys611Tyr: 37.5% (3/8)p.Cys618Phe: 0%p.Leu790Phe: 0%  **Single-gland disease:** p.Cys634Arg (2/6), p.Cys611Tyr (3/6), p.Cys618Phe  **Multi-glandular disease**: p.Cys634Arg (2/6), p.Cys611Tyr (1/6), p.Leu790Phe
[39]	 **Prevalence of PHPT, according to pathogenic variant risk category:**  **high:**N1: 23.4% (11/47)N2: 8.06% (15/186)  **moderate-high:**N1: 5.12% (2/39)N2: 1.56% (2/128)  **low-moderate:**N1: 1.42% (1/70)N2: (0/132)  **Mean (95% CI) age at PTx, according to pathogenic variant risk category**  **high:**N1: 39.1 (31.7–46.7) y N2: 34.7 (27.2–42.3) y*p* = 0.370  **moderate-high:**N1: 49 y N2: 48.5 (4.0–93.0) y*p* = 0.980  **low-moderate:**N1: 43 y
[40]	 **Prevalence of PHPT, according to pathogenic variant:**p.Cys634Arg/Gly/Phe/Ser/Trp/Tyr, insHisGluLeuCys (High risk): 11.4% (27/235)p.Cys609/611/618/620/630Arg/Gly/Phe/Ser/Thy (Moderate-high risk): 2.4% (4/235)p.Glu768Asp, p.Leu790Phe, p.Val804Leu, p.Val804Met, p.Ser891.Ala (Low risk): 0.5% (1/235)*p* < 0.001  **Median (IQR) age at PTx, according to pathogenic variant:**p.Cys634Arg/Gly/Phe/Ser/Trp/Tyr, insHisGluLeuCys (High risk): 37 (25–46) yp.Cys609/611/618/620/630Arg/Gly/Phe/Ser/Thy (Moderate-high risk): 48.5 (35.3–62.5) yp.Glu768Asp, p.Leu790Phe, p.Val804Leu, p.Val804Met, p.Ser891.Ala (Low risk): 43 y*p* = 0.270  **PHPT by pathogenic variants by year:**≤1950 vs. 1951–1960 vs. 1961–1970 vs. 1971–1980 vs. 1981–1990 vs. 1991–2000 vs. 2001–2010 vs. 2011–2020: High-risk: 42% vs. 20% vs. 23% vs. 3% vs. 3% vs. 6% vs. 0% vs. 0%, *p* < 0.001Moderate-high risk: 3% vs. 9% vs. 5% vs. 0% vs. 0% vs. 0% vs. 0% vs. 0%, *p* = 0.540Low-risk: 0% vs. 0% vs. 0% vs. 4% vs. 0% vs. 0% vs. 0% vs. 0%, *p* = 0.441**Age at PTx:** from 43.5 (38.5-54.5) to 16.5 y in high-risk pathogenic variants
[41]	 **Prevalence of PHPT, according to pathogenic variant:**p.Cys618Arg: 3.2% (1/31)p.Cys634Arg/Thr/Tyr: 33.3% (5/15)*p* = 0.010
[43]	 **Prevalence of different p.Cys634 missense pathogenic variants:**p.Cys634Arg: 37.3% (63/169)p.Cys634Tyr: 28.4% (48/169) p.Cys634Phe 18.3% (31/169) p.Cys634Ser: 8.9% (15/169)p.Cys634Gly: 7.1% (12/169)
[45]	 **Prevalence of PHPT among different mutations:**p.Met918Thr: 0%p.Cys634Phe/Gly/Arg/Ser/Trp/Tyr: 29.4% (5/17)p.Cys618Phen/Arg/Ser: 4.5% (1/22)p.Leu790Phe: 10.5% (2/19)p.Val804Met: 0%p.Ser804Ala: 0%
[47]	 **Prevalence of PHPT among different mutations:**p.Cys634Tyr: 20% (2/10)p.Cys634Arg: 40% (4/10)p.Cys611Tyr: 10% (1/10)p.Cys620Arg: 10% (1/10)p.Glu768Asp: 10% (1/10)p.Cys618Phe: 10% (1/10)  **Single-gland disease:** p.Cys634Tyr (1/2), p.Cys634Arg (2/2), p.Cys611Tyr, p.Cys620Arg, p.Glu768Asp, p.Cys618Phe  **Multi-glandular disease**: p.Cys634Tyr (1/2), p.Cys634Arg (2/2)  **Symptomatic PHPT:** p.Cys634Tyr (2/2), p.Cys634Arg (4/4), p.Cys611Tyr, p.Cys620Arg, p.Glu768Asp, p.Cys618Phe
[48]	 **Prevalence of different p.Cys634 missense pathogenic variants:**p.Cys634Arg: 39.9% (85/213)p.Cys634Tyr: 25.4% (54/213)p.Cys634Phe: 25.4% (54/213)p.Cys634Gly: 8% (17/213)p.Cys634Ser: 7.5% (16/213)p.Cys634Trp: 0.5% (1/213)p.Cys634HisGluLeuCys: 2.3% (5/213)

Abbreviations: * the studied subgroups of analysis were described in Table 1; CI = confidence interval; F = female; IQR = interquartile range; M = male; MEN2 = multiple endocrine neoplasia type 2; MTC = medullary thyroid carcinoma; N = number of patients; PHPT = primary hyperparathyroidism; PTx = parathyroidectomy; SD = standard deviation; vs. = versus; y = years.

**Table 3 diseases-13-00098-t003:** Clinical picture related to the diagnosis of primary hyperparathyroidism in patients confirmed with MEN2 [36,37,38,46,47].

Reference	Main Findings Regarding PHPT Symptoms *
[36]	Nephrolithiasis: 18.2% (2/11) Osteopenia/Osteoporosis: 9.1% (1/11) Chronic kidney disease: 9.1% (1/11)
[37]	Nephrolithiasis: 0% Osteopenia/Osteoporosis: 0% Non-specific symptoms: 20% (1/5)
[38]	Asymptomatic: 75% (12/16) Osteoporosis: 12.5% (2/16) Polydipsia: 12.5% (2/16) Nephrolithiasis Polyuria Nausea **Prevalence of symptomatic PHPT by pathogenic variant:** p.Cys634Arg: 16.67% (1/6) p.Cys611Tyr: 37.5% (3/8) p.Cys618Phe: 0% p.Leu790Phe: 0%
[46]	Nephrolithiasis: 28.57% (2/7) Pancreatitis: 14.28% (1/7)
[47]	Nephrolithiasis: 80% (8/10) Polyuria: 10% (1/10) Synchronous MTC in 80% (8/10) + Metachronus MTC in 10% (1/10)

Abbreviations: * the studied subgroups of analysis were described in Table 1; F = female; IQR = interquartile range.

**Table 4 diseases-13-00098-t004:** Parathyroidectomy-related findings in surgery candidates amid the confirmation of MEN2-associated primary hyperparathyroidism [36,37,38,39,40,42,44,46,47,48].

Reference	Prevalence of Surgical Treatment and Age at PTx *	Surgical Technique	Complications and Outcome
[36]	Underwent PTx: 72.7% (8/11)	Selective resection: 100% (8/8)	PHPT persistence: 0% PHPT recurrence: 12.5% (1/8)
[37]	Underwent PTx: 100% (5/5)	Bilateral neck exploration: 100% (5/5)	NA
[38]	Underwent PTx: 81% (13/16)	Subtotal PTx: 69% (9/13), out of which subtotal PTx with auto-transplantation in 3/9 Selective PTx: 23% (3/13) Bilateral neck exploration: 8% (1/13)	PHPT persistence: 8% (1/13) PHPT recurrence: 23% (3/13) Permanent hypoparathyroidism: 46% (6/13) Laryngeal nerve palsy: 0%
[39]	Underwent PTx: 100% N1 vs. N2: Mean (95% CI) age at PTx: 40.9 (33.7–48.0) vs. 36.4 (29.3–43.4), *p* = 0.348 Low-moderate risk pathogenic variants N1: 43 y Moderate-high risk pathogenic variants Mean (95% CI): N1: 49 y N2: 48.5 (4.0–93.0) y *p* = 0.980 High-risk pathogenic variants Mean (95% CI): N1: 39.1 (31.7–46.7) y N2: 34.7 (27.2–42.3) y *p* = 0.370	NA	NA
[40]	Median (IQR) age at PTx according to pathogenic variant p.Cys634Arg/Gly/Phe/Ser/Trp/Tyr, insHisGluLeuCys (High risk): 37 (25–46) y p.Cys609/611/618/620/630Arg/Gly/Phe/Ser/Thy (Moderate-high risk): 48.5 (35.3–62.5) y p.Glu768Asp, p.Leu790Phe, p.Val804Leu, p.Val804Met, p.Ser891.Ala (Low risk): 43 y *p* = 0.270 Age at PTx: from 43.5 (38.5–54.5) to 16.5 y in high-risk pathogenic variants	NA	NA
[42]	Underwent PTx: 100% (4/4)		
[44]	Younger age at PTx in N2 vs. N1, P_log-rank_ = 0.018	NA	NA
[46]	Underwent PTx: 85.7% (6/7)	Selective resection: 66.7% (4/6) Multi-glandular resection: 33.3% (2/6)	NA
[47]	Underwent PTx: 100% (10/10)	NA	NA
[48]	Age at PTx, median (IQR) = 39 (26–46) Age at PTx, median (IQR) 1922–1950 vs. 1951–1960 vs. 1961–1970 vs. 1971–1980 vs. 1981–1990 vs. 1991–2000 vs. 2001–2010: 46 (39.5–55) vs. 42 (31–45.5) vs. 31 (23–36) vs. 26 (26–26) vs. 12 (12–12) y, *p* = 0.008	NA	PHPT persistence: 16.7% (1/6) Permanent hypoparathyroidism: 33.3% (2/6) Transient hypoparathyroidism: 33.3% (2/6) Permanent unilateral vocal cord palsy: 16.7% (1/6)

Abbreviations: * the studied subgroups of analysis were described in Table 1; CI = confidence interval; F = female; IQR = interquartile range; M = male; MEN2 = multiple endocrine neoplasia type 2; N = number of patients; NA = not available; PHPT = primary hyperparathyroidism; PTx = parathyroidectomy; vs. = versus; y = years.

**Table 5 diseases-13-00098-t005:** Histological features in patients with primary hyperparathyroidism and MEN2 [36,37,38,39,42,46,47].

Reference	Pathology Findings *
[36]	Uni-glandular disease: 87.5% (7/8) Multi-glandular disease: 12.5% (1/8)
[37]	Uni-glandular disease: 75% (3/5) Multi-glandular disease: 25% (1/5) Size, median (IQR): 0.7 (0.55–0.9) cm Mass, median (IQR): 118 (56.3–302) mg
[38]	Uni-glandular disease: 50% (6/13) Multi-glandular disease: 42% (6/13)
[39]	N1 vs. N2: Primary tumor: mean (95% CI) diameter: 37.2 (29.8–44.7) vs. 40.7 (33.9–47.5) mm, *p* = 0.505
[42]	Uni-glandular disease: 50% (2/4) Multi-glandular disease: 50% (2/4)
[46]	Uni-glandular disease: 33.3% (2/6) Multi-glandular disease: 50% (3/6) No parathyroid tissue: 16.7% (1/6)
[47]	Uni-glandular disease: 70% (7/10) Multi-glandular disease: 30% (3/10)

Abbreviations: * the studied subgroups of analysis were described in Table 1; CI = confidence interval; IQR = interquartile range; N = number of patients.

**Table 6 diseases-13-00098-t006:** Findings regarding medullary thyroid carcinoma in MEN2 patients (the data according to the studies that also provided an analysis of the primary hyperparathyroidism) [35,38,39,40,41,43,44,46,47,48].

Reference	Main Findings *
[35]	**Prevalence of MTC, according to pathogenic variant:** p.Met918Thr: 100% (2/2) p.Cys634Phe/Gly/Arg/Ser/Trp/Tyr: 88.9% (224/27) p.Gly533Cys: 50% (1/2) p.Cys611Phe/Gly/Ser/Tyr/Trp: 52% (13/25) p.Cys618Phe/Arg/Ser/Tyr: 90% (9/10) p.Cys620Phe/Arg/Ser/Tyr: 87.5% (7/8) p.Cys630Arg/Tyr: 100% (1/1) p.Glu768Asp: 80% (4/5) p.Leu790Phe: 77.8% (7/9) p.Val804Met/Leu: 51.8% (29/56) p.Ser891Arg: 92.3% (12/13) **Median (min, max) age at MTC diagnosis, according to pathogenic variant:** p.Met918Thr: 15.5 (8, 23) y p.Cys634Phe/Gly/Arg/Ser/Trp/Tyr: 24.5 (4, 72) y p.Gly533Cys: 29 y p.Cys611Phe/Gly/Ser/Tyr/Trp: 50.5 (34, 79) y p.Cys618Phe/Arg/Ser/Tyr: 38 (27, 59) y p.Cys620Phe/Arg/Ser/Tyr: 36 (21, 55) y p.Cys630Arg/Tyr: 58 y p.Glu768Asp: 48 (39, 68) y p.Leu790Phe: 55 (10, 81) y p.Val804Met/Leu: 56 (16, 77) y p.Ser891Arg: 50 (15, 75) y
[38]	**Thyroidectomy in 100% of N1 (16/16):** MTC 69% (11/16) + C-cell hyperplasia 31% (5/16) **MTC diagnosis:** before PHPT 45% (5/11) + synchronous with PHPT: 55% (6/11)
[39]	**N1 vs. N2:** **MTC prevalence:** 97.4% vs. 57.0%, *p* < 0.001 **Age at thyroidectomy**: mean (95% CI): 45.4 (42.8–47.9) vs. 30.5 (28.1–32.8) y, *p* < 0.001 **Largest primary tumor**: mean (95% CI): 19.5 (16.8–22.1) vs. 7.9 (6.6–9.1) mm, *p* < 0.001 **Carriers with node-positive MTC**: 71.5% vs. 29.5%, *p* < 0.001 **Carriers with biochemical cure**: 34.1% vs. 74.8%, *p* < 0.001
[40]	**High risk vs. moderate-high vs. low-moderate:** **MTC prevalence**: 75% vs. 65.2% vs. 63.2%, *p* = 0.016 **Age at thyroidectomy:** median (IQR): 17 (6–31) vs. 29 (9–42) vs. 39 (23–56), *p* < 0.001
[41]	p.Cys618Arg vs. p.Cys634Arg/Thr/Tyr: **Tumor size:** mean ± SD: 8.9 ± 6.7 vs. 18.5 ± 11.1 mm, *p* = 0.004 **Preoperative calcitonin level, proportion of upper limit:** 84.5 ± 201.9 vs. 333.9 ± 314.5, *p* = 0.030
[43]	**N1 vs. N2:** **Age at thyroidectomy:** median (IQR): 8 (4–23) vs. 12 (6–25) y, *p* = 0.145 **MTC prevalence**: 68% vs. 70%, *p* = 0.869 **Node metastases**: 19% vs. 45%, *p* = 0.006
[44]	**N1 vs. N2:** **MTC prevalence:** 59% vs. 56%, *p* = 0.609 **Age at thyroidectomy:** median (IQR): 19 (6–33) vs. 17 (6.5–32) y, *p* = 0.705 **Node metastases:** 29% vs. 43%, *p* = 0.029
[46]	**N1 vs. N2 vs. N4 vs. N5:** **Age at MTC diagnosis:** mean ± SD: 30.15 ± 15.3 vs. 40.09 ± 14.78 vs. 35.3 ± 11.9 vs. 28.3 ± 19.8 N1 vs. N2: *p* < 0.05 N1 vs. N5: *p* < 0.05 **Size of thyroid nodule**: mean ± SD: 2.44 ± 1.35 vs. 3.14 ± 1.43 vs. 2.96 ± 1.38 vs. 2.9 ± 0.85 N1 vs. N5: *p* < 0.05 N2 vs. N5: *p* < 0.05 N4 vs. N5: *p* < 0.05 **Cured after surgery**: 46% vs. 37% vs. 35% vs. 91%; N1 vs. N5: *p* < 0.05; N2 vs. N5: *p* < 0.05; N4 vs. N5: *p* < 0.05
[47]	**MTC** in 100% (10/10)
[48]	**MTC**: 76.5% (163/213) **C-cell hyperplasia**: 21.6% (46/213) **Age at thyroidectomy:** median (IQR): 17 (6031.5) y **Largest tumor diameter**: median (IQR): 6 (3–18) mm **Node metastases**: median (IQR): 0 (0–3)

Abbreviations: * the studied subgroups of analysis were described in Table 1; CI = confidence interval; F = female; IQR = interquartile range; M = male; MTC = medullary thyroid carcinoma; N = number of patients; PC = pheochromocytoma; PTx = parathyroidectomy; y = years; vs.= versus (bold font means the parameters that has been analyzed).

**Table 7 diseases-13-00098-t007:** Findings regarding pheochromocytoma in MEN2 patients (the data according to the studies that also provided an analysis of the primary hyperparathyroidism) [35,38,39,40,41,43,44,45,46,47,48].

Reference	Main Findings *
[35]	**Prevalence of PC, according to pathogenic variant:** p.Cys634Phe/Gly/Arg/Ser/Trp/Tyr: 55.6% (15/27) p.Met918Thr: 50% (1/2) p.Cys611Phe/Gly/Arg/Ser/Tyr: 28% (7/25) p.Cys618Phe/Arg/Ser/Tyr: 20% (2/10) p.Cys620Phe/Arg/Ser/Tyr: 25% (2/8) p.Leu790Phe: 11.1% (1/9) p.Ser891Arg: 7.7% (1/13) **Median (min, max) age at AD, according to pathogenic variant:** p.Cys634Phe/Gly/Arg/Ser/Trp/Tyr: 29 (18, 72) y p.Met918Thr: 31 y p.Cys611Phe/Gly/Arg/Ser/Tyr: 55 (29, 86) y p.Cys618Phe/Arg/Ser/Tyr: 47.5 (37, 58) y p.Cys620Phe/Arg/Ser/Tyr: 43 (32, 54) y p.Leu790Phe: 62 y p.Ser891Arg: 75 y
[38]	**PC prevalence in N1**: 56% (9/16): synchronous: 56% (5/9) + before PHPT: 22% (2/9) + after PHPT: 22% (2/9)
[39]	**N1 vs. N2:** **PC prevalence**: 30.1% vs. 13.2%, *p* < 0.001 **Age at AD**: mean (95% CI) = 37.5 (34.1–41.0) vs. 40.7 (33.9–47.5) y, *p* = 0.431
[40]	**High risk vs. moderate-high vs. low-moderate:** **PC prevalence**: 32.1% vs. 16.4% vs. 3%, *p* < 0.001 **Age at AD**: median (IQR): 34 (26–42) vs. 40 (32–48) vs. 32.5 (28.5–46), *p* = 0.118 **Contralateral PC**: 19.4% vs. 4.8% vs. 1.5%, *p* < 0.001
[41]	**N1 vs. N2**: PC was found in 6.5% vs. 53.3%, *p* = 0.001
[43]	**N1 vs. N2:** **PC prevalence**: 19% vs. 33%, *p* = 0.051 **Bilateral PC**: 10% vs. 24%, *p* = 0.021
[44]	**N1 vs. N2**: **PC prevalence:** 13% vs. 19%, *p* = 0.094 **Bilateral PC**: 5% vs. 12%, *p* = 0.005 **First PC inheritance from father:** HR (95% CI) = 1.8 (1.1–3.0) *p* = 0.020
[45]	**Prevalence of PC by pathogenic variant:** p.Cys618Phe/Arg/Ser: 4.5% (1/22) p.Cys634Phe/Gly/Arg/Ser/Trp/Tyr: 70.6% (12/17) p.Leu790Phe: 0% p.Val804Met: 0% p.Ser891Ala: 0% p.Met918Thr: 0%
[46]	**N3: Prevalence PC** of 48.64% (18/37)
[47]	**Prevalence of PC**: 70% (7/10) **Bilateral PC**: 42.85% (3/7) + **Unilateral PC**: 57.15% (4/7)
[48]	**PC in first adrenal gland**: **Carriers with PC**: 31.0% (66/213) → **Age at AD**: median (IQR): 34 (26–42) y **PC in second adrenal gland**: 18.8% (40/213) → **Age at AD**: median (IQR): 35 (29–41.8) y

Abbreviations: * the studied subgroups of analysis were described in Table 1; AD = adrenalectomy; CI = confidence interval; F = female; IQR = interquartile range; M = male; MTC = medullary thyroid carcinoma; N = number of patients; PC = pheochromocytoma; PTx = parathyroidectomy; y = years (bold font means the parameters that has been analyzed).

**Table 8 diseases-13-00098-t008:** Case reports of PHPT in MEN2 patients (the display starts with the most recent publication date) [136,137,138,139,140].

First Author Publication Year Reference Number	Studied Population	*RET* Pathogenic Variant	Clinical Picture and Family Medical History	Treatment and Outcome
La Greca 2024 [136]	Female, 40 y	p.K666N	PHPT Pheochromocytoma—right adrenal mass of 4.5 cm × 4.2 cm Plasma metanephrines = 1957 pg/mL (normal: <57) Plasma normetanephrines = 1329 pg/mL (normal: <148) MTC Calcitonin = 12.3–30.7 mg/dL (normal: 0–5.1) No family history Pathogenic variant identified in sister, daughter, and brother	Adrenalectomy Total thyroidectomy and selective PTx
Jones 2024 [137]	Female, 26 y	NA (unspecified *RET* pathogenic variant)	Asymptomatic PHPT—adenoma and ectopic (mediastinal) parathyroid gland Multifocal MTC Bilateral pheochromocytoma No family history	PTx and total thyroidectomy, followed by bilateral adrenalectomy and thoracoscopic parathyroidectomy and thymectomy → postoperative hypoparathyroidism
Kim 2022 [138]	Female, 64 y	p.T244I (unknown significance)	Right hip pain at onset History of PHPT—uni-glandular disease Pancreatic retroperitoneal paraganglioma of 11 cm No family history	PTx
Brown 2020 [139]	female, 28 y	Cys630Tyr and p.Ala176Leufs*10 (loss of function MEN1 pathogenic variant)	Screening Low bone mass for age PHPT MTC Family history: PHPT: sister MTC: paternal great aunt Zollinger–Ellison syndrome: sister, father, paternal uncle Thymus carcinoid: father	Total PTx with left forearm re-implantation → postoperative hypoparathyroidism Total thyroidectomy with regional lymph node dissection Partial thymectomy
Giani 2020 [140]	Female, 7 y	p. Asp631_Leu633delinsGlu, de novo	Marfanoid habitus Bilateral mucosal neuromas of the mouth History of plexiform neurofibroma and ganglioneuromatosis PHPT MTC No family history	Total thyroidectomy Bilateral neck exploration for PHPT

Abbreviations: MTC = medullary thyroid carcinoma; PTH = parathormone; PHPT = primary hyperparathyroidism; PTx = parathyroidectomy; y = year.

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
