# Peer review of "An Analysis of Primary Hyperparathyroidism in Individuals Diagnosed with Multiple Endocrine Neoplasia Type 2"

_diseases, 2025, doi:10.3390/diseases13040098_

Round 1
Reviewer 1 Report
Comments and Suggestions for Authors
This is a narrative review of 14 papers concerning PHPT in MEN 2 patients, analysing 218 patients published by other authors. It may be of interest because it nicely summarizes clinical and genetic data of this particular condition.
The methodology of a narrative review is of very limited scientific value, but this was the original intention of the authors and cannot be modified.
Author Response
Response to Review 1 Comments
Dear Reviewer,
Thank you very much for your time and your effort to review our manuscript.
We are very grateful for providing your valuable feedback on the article.
Here is our response and related amendment that has been made in the manuscript according to your review (marked in yellow color).
This is a narrative review of 14 papers concerning PHPT in MEN 2 patients, analyzing 218 patients published by other authors. It may be of interest because it nicely summarizes clinical and genetic data of this particular condition.
Thank you very much. We really appreciate it!
The methodology of a narrative review is of very limited scientific value, but this was the original intention of the authors and cannot be modified.
Thank you very much. We confirm and we mentioned it within the main text of this comprehensive analysis. Thank you
Thank you very much.

Reviewer 2 Report
Comments and Suggestions for Authors
The manuscript by Gheorghe et al. is a comprehensive review about the clinical features of primary hyperparathyroidism in patients affected with MEN2A. The review is well written, methodology is stated and appropriated, available studies are adequately analyzed.
I suggest the Authors to highlighting the main finding on management of primary hyperparathyroidism in MEN2A patients providing a flow chart based on the evidence from the literature.
Comments on the Quality of English LanguageThe manuscript is readable, however English language should be carefully revised in order to avoid some unusual terms.
Author Response
Response to Review 2 Comments
Dear Reviewer,
Thank you very much for your time and your effort to review our manuscript.
We are very grateful for your insightful comments and observations, also, for providing your valuable feedback on the article.
Here is a point-by-point response and related amendments that have been made in the manuscript according to your review (marked in yellow color).
Suggestions for Authors
The manuscript by Gheorghe et al. is a comprehensive review about the clinical features of primary hyperparathyroidism in patients affected with MEN2A. The review is well written, methodology is stated and appropriated, available studies are adequately analyzed.
Thank you very much. We really appreciate it!
I suggest the Authors to highlighting the main finding on management of primary hyperparathyroidism in MEN2A patients providing a flow chart based on the evidence from the literature.
Thank you very much. According to your recommendation, we introduced a flow chart with main findings (Figure 2). Thank you
Comments on the Quality of English Language: The manuscript is readable, however English language should be carefully revised in order to avoid some unusual terms.
Thank you very much. According to your recommendation, we revised the English language. Thank you
Thank you very much.

Reviewer 3 Report
Comments and Suggestions for Authors
Wonderful review
You should describe about the highest serum calcium level----to reveal the severity in MEN2-HPT
In the discussion: Discuss more about the balance between aggressive surgical procedure and postoperative complication, and the role of cryopreservation, intra-operative PTH assay, and ICG perfusion in the special PHPT Operation
Author Response
Response to Review 3 Comments
Dear Reviewer,
Thank you very much for your time and your effort to review our manuscript.
We are very grateful for your insightful comments and observations, also, for providing your valuable feedback on the article.
Here is a point-by-point response and related amendments that have been made in the manuscript according to your review (marked in yellow color).
Wonderful review
Thank you very much. We really appreciate it!
You should describe about the highest serum calcium level to reveal the severity in MEN2-HPT.
Thank you very much. According to your interesting recommendation, we expanded this aspect at Discussion (e.g….”The importance of PHPT diagnosis in MEN2 patients should not be downplayed by the relatively less severe presentation compared with other syndromic manifestations such as the presence of a potentially severe thyroid malignancy, especially if PHPT co-occurs with MTC, making preoperative diagnosis of both conditions crucial for the surgical approach [72,73]. Notably, a case-control analysis of the serum calcium levels in PHPT amid MEN2 confirmation (in order to highlight the impact of the acute hypercalcemia and a potential more sever presentation) was not provided by all the studies we could identify [35,38-43] or the results only included data with a relative low statistical power [37]. On the other hand, as mentioned, Figueiredo et al. [36] showed that serum total calcium, respectively, PTH were statistically significant lower in MEN2 than MEN1, of 10.6 10.6 ± 1.1 versus 11.7 ± 1.2 mg/dL (p = 0.03), respectively, 108 versus 196.9 pg/mL (p = 0.01), but, also, than found in hyperparathyroidism-jaw tumour syndrome, of 12.9 ± 1.8 mg/dL (p < 0.001), respectively, 383.5 pg/mL (p = 0.01) [36].” Thank you
In the discussion: Discuss more about the balance between aggressive surgical procedure and postoperative complication, and the role of cryopreservation, intra-operative PTH assay, and ICG perfusion in the special PHPT Operation.
Thank you very much. According to your recommendation, we expanded these aspects if the sample-focused analysis supported them (e.g. …”nowadays, if parathyroid glands are macroscopically normal, we consider that they should not be removed, noting the low rate of recurrence and of surgical complications... Currently, a less aggressive and minimally invasive approach is preferred in order to reducing the post-operatory rate of complications, including hypocalcemia and hypoparathyroidism... The use of intra-operatory PTH assays helps the rate of complete gland/tumour removal and avoids unnecessary redo surgery...If total PTx with auto-transplantation is performed, regular follow-up is still needed, due to PHPT recurrence or multi-gland involvement, including in ectopic parathyroid... As mentioned, we identified a single study that explored intraoperative auto-fluorescence [42], a novel technique used for identifying parathyroid glands based on the natural fluorescence emitted by the parathyroid tissue during exposure to near-infrared light (which is especially useful in bilateral neck exploration) [131-134]. The clinical and therapeutic implications in MEN2 patients, however, remain an emerging topic that needs further studies. No additional data with respect to using intraoperative indocyanine green angiography for glands localization in MEN2 we could identify, neither in using cryopreservation, but these seem promising alternatives for selected cases amid a tailored multimodal management in MEN2-related PHPT.” Thank you
Thank you very much.
